# Essential Oil of *Mentha aquatica var. Kenting Water Mint* Suppresses Two-Stage Skin Carcinogenesis Accelerated by BRAF Inhibitor Vemurafenib

**DOI:** 10.3390/molecules24122344

**Published:** 2019-06-25

**Authors:** Chih-Ting Chang, Wen-Ni Soo, Yu-Hsin Chen, Lie-Fen Shyur

**Affiliations:** 1Agricultural Biotechnology Research Center, Academia Sinica, Taipei 115, Taiwan; 2Department of Biological Science and Technology, National Taiwan University, Taipei 106, Taiwan; 3Taichung District Agricultural Research and Extension Station, Council of Agriculture, Executive Yuan, Taichung 515, Taiwan; 4Graduate Institute of Pharmacognosy, Taipei Medical University, Taipei 110, Taiwan

**Keywords:** BRAF inhibitor, *Mentha aquatica var. Kenting Water Mint*, essential oil, chemoprevention, two-stage skin carcinogenesis

## Abstract

The v-raf murine sarcoma viral homolog B1 (BRAF) inhibitor drug vemurafenib (PLX4032) is used to treat melanoma; however, epidemiological evidence reveals that it could cause cutaneous keratoacanthomas and squamous cell carcinoma in cancer patients with the most prevalent *HRAS^Q61L^* mutation. In a two-stage skin carcinogenesis mouse model, the skin papillomas induced by 7,12-dimethylbenz[a]anthracene (DMBA)/12-*O*-tetradecanoylphorbol-13-acetate (TPA) (DT) resemble the lesions in BRAF inhibitor-treated patients. In this study, we investigated the bioactivity of *Mentha aquatica var. Kenting Water Mint* essential oil (KWM-EO) against PDV cells, mouse keratinocytes bearing *HRAS^Q61L^* mutation, and its effect on inhibiting papilloma formation in a two-stage skin carcinogenesis mouse model with or without PLX4032 co-treatment. Our results revealed that KWM-EO effectively attenuated cell viability, colony formation, and the invasive and migratory abilities of PDV cells. Induction of G_2_/M cell-cycle arrest and apoptosis in PDV cells was also observed. KWM-EO treatment significantly decreased the formation of cutaneous papilloma further induced by PLX4032 in DT mice (DTP). Immunohistochemistry analyses showed overexpression of keratin14 and COX-2 in DT and DTP skin were profoundly suppressed by KWM-EO treatment. This study demonstrates that KWM-EO has chemopreventive effects against PLX4032-induced cutaneous side-effects in a DMBA/TPA-induced two-stage carcinogenesis model and will be worth further exploration for possible application in melanoma patients.

## 1. Introduction

Cutaneous squamous cell carcinoma (cuSCC) and keratoacanthoma (KA) develop in approximately 20% to 30% of patients who are treated with BRAF (v-raf murine sarcoma viral homolog B1) inhibitors, such as vemurafenib (PLX4032) [1]. Functional studies have demonstrated that these serious side-effects caused during the treatment of PLX4032 are through paradoxical activation of the MAPK signaling pathway of wild-type BRAF cell lines bearing either oncogenic *RAS* mutations or upstream receptor tyrosine kinase activity [2,3,4]. In a recent study, cuSCC and KAs emerging from patients administrated with BRAF inhibitor were analyzed for oncogenic mutations and activating mutations on *RAS*, especially the *HRAS* isoform was noticed in about 60% of subjects [5]. Among the *RAS* mutants, *HRAS^Q61L^* was the most prevalent, and thus, the genetic *HRAS^Q61L^* mutation of cells (e.g., keratinocytes PDV) was selected to investigate the pre-clinical pathological mechanisms [6]. Meanwhile, the mouse skin model of multiple-stage chemical carcinogenesis is a representable in vivo model for understanding the development of cuSCC [7,8]. Topical exposure of carcinogens, 7,12-dimethyl[a]anthracene (DMBA), as a tumor initiator results in *HRAS^Q61L^* mutation in mouse skin. Subsequently, topical treatment of tumor promoter, 12-*O*-tetradecanoyl-phorbol-13-acetate (TPA) then leads to the formation of lesions, KAs, and the development of SCC. FVB (Friend Virus B NIH Jackson) mice administrated with DMBA/TPA along with BRAF inhibitor, PLX4720, showed a remarkable acceleration in the appearance of lesions, an increase of incidence, and enhanced progression to KAs and SCC which resemble the papillomas induced by BRAF inhibitors in the clinical setting [5].

Tumor development is correlated with proliferation and expansion of not only cancer cells but also stroma, vessels, and infiltrating inflammatory cells and elements [9]. Neoplastic growth is related to a prolonged inflammatory condition induced by extrinsic or intrinsic pathways. The extrinsic pathways are related to a continued inflammatory condition, while the intrinsic pathways are stimulated by genetic transformations, which result in the activation of oncogenes or inactivation of tumor suppressor genes [10]. Cells with an altered phenotype propagate the secretion of inflammatory mediators, thus triggering the formation of a tumor microenvironment (TME) and development of tumors [11]. Recently, immunoinflammatory cells, such as macrophages, have been identified as critical contributors to malignancies in various tumor types, such as melanoma, lung carcinoma, glioma, gastric cancer, and wound-induced skin cancer [12,13]. 

Numerous studies have demonstrated that essential oils (EOs) of *Mentha* species have antiviral, antimicrobial, antioxidant, anti-inflammatory, and anti-tumor activities [14,15,16,17,18]. The objective of this study was to investigate the bioefficacy of EO from *Mentha aquatica var. citrata Kenting Water Mint* (KWM-EO) against two-stage skin carcinogenesis, with or without PLX4032 irritation, and the underlying molecular mechanisms. The chemical components of KWM-EO were analyzed using GC×GC-TOF MS, and its effect on *HRAS* mutant PDV keratinocyte activity was further investigated. Our in vitro bioassay results demonstrated that KWM-EO treatment suppressed PDV cell viability, colony formation ability, and induced G_2_/M cell-cycle arrest and cell apoptosis in the presence and absence of PLX4032. KWM-EO also inhibited proinflammatory cell infiltration and papilloma formation in DMBA/TPA-induced two-stage skin carcinogenesis facilitated by PLX4032 in mice.

## 2. Results

### 2.1. Chemical Compositions of Mentha aquatica var. Kenting Water Mint Essential Oil

KWM-EO was obtained by hydrodistillation of the aerial parts. The chemical profile of KWM-EO was analyzed by GC×GC-TOF MS. Twenty compounds representing 81.86% of the total content were identified in KWM-EO (Table 1). Monoterpene hydrocarbons accounted for 56.01% of KWM-EO with 22.18% *β*-ocimene as the most abundant component, and *β*-pinene and *α*-pinene accounting for 15.41% and 10.49%, respectively. KWM-EO was identified to contain 15.86% oxygenated monoterpenes, of which eucalyptol (12.87%) was the most abundant.

### 2.2. KWM-EO Effect on PDV Cell Proliferation, Invasion, and Migration

The PDV cell line is a mouse keratinocyte bearing *HRAS^Q61L^* mutation, which is the most relevant mutation in BRAF inhibitor-induced cutaneous squamous cell carcinoma. The PDV cell viability after treatment with 0 to 100 μg/mL KWM-EO was determined by MTT assay. The cell viability was decreased when KWM-EO concentration increased. When the PDV cells were treated with up to 100 μg/mL of KWM-EO for 24 h, the cell viability was inhibited to 53.31% (Figure 1A). The long-term colony formation ability of PDV cells was determined by treating with KWM-EO alone or in the presence of PLX4032 (PLX). The MEK (mitogen-activated protein kinase kinase) inhibitor, selumetinib (AZD6244), was used as a reference control. As shown in Figure 1B, 0.5 μM PLX4032 treatment promoted the colony formation of PDV cells compared to the vehicle-treated cells. In the presence or absence of PLX4032, KWM-EO treatment showed a dose-dependent effect, and KWM-EO treatment at the high dose of 40 μg/mL revealed a better effect than 0.5 μM AZD6244 treatment. PDV cell invasive ability was investigated by Matrigel coated-transwell assay. The result showed that PLX4032 treatment facilitated cell invasion relative to vehicle treatment, and KWM-EO suppressed the invasive ability on concentration-dependence (Figure 1C). In wound healing assay representing cell migratory ability, 2 μM PLX4032 treatment significantly and time-dependently increased cell migration. The migratory ability of PDV cells was restricted by 50 μg/mL KWM-EO treatment with or without PLX4032 stimulation (Figure 1D).

### 2.3. KWM-EO Induces Cell-Cycle Arrest and Apoptosis in PDV Cells

The KWM-EO effect on the PDV cell-cycle machinery was determined using flow cytometry. The analysis demonstrated that PLX4032 treatment alone had no significant effect on the cell-cycle of PDV cells; however, the cell-cycle profile treated with KWM-EO exhibited G_2_/M arrest. After KWM-EO treatment for 24 h, the percentage of cells in the G_2_/M phase was raised from 33.0–33.6% to 44.3–45.0%, in the presence or absence of PLX4032 (Figure 2A). According to cell-cycle analysis, an elevated percentage of the sub-G_1_ population was also observed with KWM-EO and PLX4032+KWM-EO treatment. Thus cell apoptosis was further examined. Cells were stained with annexin V and propidium iodide and analyzed by flow cytometry. The data demonstrated that treatment with 75 μg/mL KWM-EO strongly induced 86.9% and 80.7% apoptotic cells in the presence or absence of PLX4032, respectively (Figure 2B). Western blotting was further used to explore the protein expression profile related to G_2_/M cell-cycle arrest and cell apoptosis. Cyclin B1-cell division cycle protein 2 (cdc2), also known as M-phase promoting factor (MPF), regulates G_2_/M transition. Phosphorylation of Thr161 in cdc2 is required for activation of MPF and brings on the onset of mitosis [19]. Treatment with 75 μg/mL KWM-EO for 24 h reduced the protein expression level of cyclin B1 and p-cdc 2 (Thr161), suggesting the inhibition of cell mitosis. Phosphorylation of cdc25C, which is responsible for activation of MPF, was also decreased (Figure 2C). The initiation of apoptotic cell death is needed to activate a group of intracellular cysteine proteases, named caspases. The cleavage of poly(ADP-ribose) polymerase-1 (PARP-1) by caspases is regarded to be a characteristic of cell apoptosis [20]. After treatment with 75 μg/mL KWM-EO for 12 h, both hallmarks of apoptosis, caspase 3 and PARP-1, were cleaved into their activated forms (Figure 2D). Paradoxical MAPK activation is known to be the main reason for cutaneous squamous cell carcinoma induced by BRAF inhibitor in *RAS* mutant cells [21]. KWM-EO (75 μg/mL) treatment for 24 h significantly inhibited ERK and p-ERK expression in PDV cells; while in treatment with 0.5 μM PLX4032, the re-activation of p-MEK and p-ERK was observed, which could be reversed by KWM-EO and MEK inhibitor (Figure 2E). 

### 2.4. KWM-EO Inhibits Two-Stage Skin Carcinogenesis in FVB Mice

To investigate the chemopreventive effect of KWM-EO in BRAF inhibitor-induced cutaneous squamous cell carcinoma, a DMBA-initiated and TPA-promoted two-stage skin carcinogenesis model was established, and the study diagram is shown in Appendix A. After topical application of 25 μg DMBA and 4 μg TPA (DT) on mouse dorsal skin for 12 weeks, papillomas were successfully induced. If mice were co-treated with 20 mg/kg/BW PLX4032 (DTP), starting at 6 weeks, bigger and more papillomas occurred (Figure 3A). The DT and DTP groups developed papillomas on the skin as early as 5 weeks after TPA treatment. Within 6 to 7 weeks, the tumor incidence in DTP mice was much higher than in DT mice; at 8 weeks, the tumor incidence in the DT and DTP group reached 100%, while KWM-EO treatment delayed and decreased the tumor incidence (Figure 3B). On average, the DT group developed 15.4 papillomas/mouse at 12 weeks, the topical application of 5 mg KWM-EO reduced the average number of papillomas to 7.9/mouse. Under stimulation of PLX4032 in DTP mice, the average number of papillomas was raised to 22.4/mouse which was ameliorated by KWM-EO to 9.1 papillomas/mouse at 12 weeks (*P* < 0.05) (Figure 3C). The dot histogram shows the distribution of papilloma number per mouse and the median in a group (Figure 3D). The papilloma number was significantly decreased by KWM-EO treatment in DTP mice. Mouse body weights were recorded every week during the experimental period, and the results show that the body weights in all treated mice were similar to the sham control mice (Appendix A). To examine the toxicology of these applied compounds and essential oil, mouse organ index was calculated, and H&E staining was executed to observe the organ structure and pathology. The mouse organ index was unchanged for the heart, lung, and kidney within the groups; however, the index of the liver organ was lower in the KWM-EO-treated mice, and the index of the spleen organ was increased in PLX4032-treated mice. The H&E staining result on the organs showed that there were no observable differences between the sham and all the treatment groups (Figure 3F).

### 2.5. Skin Histology and Epidermal Cell Proliferation in KWM-EO-Treated Mice

Effect of topically applied KWM-EO on chemically induced skin tumorigenesis was further observed by skin histologic changes. The skin structure was first examined by H&E staining. With DMBA and TPA treatment in the presence or absence of PLX4032, the thickness of the epidermis was elevated compared to the sham group (Figure 4A), while the epidermal hyperplasia was attenuated by repeated treatment with KWM-EO for 12 weeks. The main cell type responsible for hyper-proliferative epidermis was further explored by immunofluorescent staining. Ki67 is a representative marker of cell proliferation, and cytokeratin 14 (K14) is the intermediate filament protein of basal keratinocytes. From the microphotographs, the expression level of ki67 showed remarkable upregulation in the DT and DTP group. After merging both ki67 and K14 staining with DAPI, the hyperplasia of the epidermis arising from basal keratinocytes was seen which was alleviated by KWM-EO topical treatment (Figure 4B). In addition, the paradoxical MAPK activation that could lead to cell proliferation in *RAS* mutant cells treated with BRAF inhibitors was also investigated. The IHC staining result revealed considerable p-ERK protein between the dermis and papilloma, especially in the group with PLX4032 stimulation; and this activation was significantly diminished by KWM-EO treatment (Figure 4C).

### 2.6. Anti-Inflammatory Effect of KWM-EO

Inflammation is a vital element in the progression of two-stage skin carcinogenesis. COX-2, a pro-inflammatory enzyme commonly observed in inflamed cells or tissues was counter-stained with pro-inflammatory immune cells, neutrophils (neutrophil elastase+) and macrophages (F4/80+). The overexpression of COX-2 was increased in both DT and DTP mouse dorsal skin. Interestingly, most of the COX-2 proteins were observed colocalized with infiltrated neutrophils, but not macrophages (Figure 5A,B). Upon treatment with KWM-EO, the neutrophil and macrophage infiltration and upregulation of COX-2 were alleviated (Figure 5A,B).

## 3. Discussion

EOs have been utilized as fragrances, food flavorings, and folk medicines, among other applications throughout human history. In recent decades, a large number of studies have reported chemical constituent analysis of EOs and investigated their bio-efficacy and the responsible bioactive compounds. EOs from *Mentha* species have been reported to have anti-inflammatory, anti-oxidant, anti-fungal, and anti-bacterial activities [14,15,16,17,18]. The ethanolic extract of *Mentha×piperita* L., a cross-species of watermint and spearmint, at 50 and 100 µg/mL, suppressed LPS-induced nitric oxide production in macrophages by 18.85% and 41.88% inhibition, respectively [22]. Anti-cancer cell activity has also been reported for mint EOs. For example, *Mentha×piperita* L. extract showed more potent activity against proliferative activity of human MDA-MB-231 breast cancer cells (cell inhibition ratio = 46.53% at 150 μg/mL) than human A375 melanoma cells (cell inhibition ratio = 25.08% at 150 μg/mL) [14]. This study is the first to investigate and observe that KWM-EO can prevent two-stage skin carcinogenesis chemically induced by DMBA/TPA and its acceleration by BRAF inhibitor drug PLX4032. The two-stage skin carcinogenesis mouse model established by DMBA and TPA irritation is considered to be a representative study system through which to explore the pathology and underlying mechanisms of human squamous cell carcinomas [23]. It has also been used to evaluate the BRAF^V600E^ inhibitor drugs, such as vemurafenib-induced cutaneous side-effects, including SCC and KA in patients [3]. We, thus, established this two-stage skin carcinogenesis mouse model, and the bioactivities of KWM-EO were examined. Our results indicated that KWM-EO treatment significantly inhibited papilloma incidence and number in DT and DTP mice. According to the histopathological analysis of skin tissue sections, KWM-EO not only attenuated the abnormal proliferation and hyperplasia of the epidermis but also decreased the inflammatory neutrophil and macrophage infiltration and COX-2 overexpression in neutrophils. Moreover, this study is the first to observe that abnormal epidermis proliferation in DT and DTP mice was mainly contributed by keratinocytes as a co-positively stained marker protein K14 and proliferation marker ki67. Topical administration of KWM-EO can reverse the over proliferation of K14 keratinocytes in DT- or DTP-irritated mouse skins. 

A previous review article published by Pandey et al. [24] summarized that EOs of some *Ocimum* species exhibited anti-inflammatory and anti-cancerous properties which contain pinene, β-ocimene, and linalool, the chemical constituents present in KWM-EO. α-Pinene present in KWM-EO by 10.49% was reported to induce cell apoptosis and disrupt mitochondrial potential in B16F10 cells, and it also effectively reduced melanoma lung metastasis [25]. β-Caryophyllene accounted for 2.8% in KWM-EO was a major compound in the EO of *P. missionis*. Pavithra et al. demonstrated that EO from *P. missionis* induced cell death through intrinsic mitochondrial and extrinsic apoptotic pathways in A431 and HaCaT cells [26]. The results from these studies might potentially support part of our observations for the anti-inflammatory and chemopreventive activities of KWM-EO against two-stage skin carcinogenesis. 

PDV keratinocytes harboring *HRAS* mutation are commonly found in DT-induced mouse SCC [27]. We adapted this cell model to investigate the in vitro effect and modes of action of KWM-EO. Our data revealed that KWM-EO treatment significantly diminished PDV cell colony formation ability and suppressed reactivation of MEK-ERK signaling stimulated by PLX4032. The PDV cell invasive and migratory abilities were promoted by PLX4032, which were suppressed by KWM-EO treatment. KWM-EO also induced G_2_/M arrest in PDV cells through deregulating p-cdc2, p-cdc25C, and cyclin B1 proteins. The cell apoptosis induced by KWM-EO was through activation of caspase 3 and PARP-1 after cells were treated for 6 h. These in vitro data support in part the inhibitory activity of KWM-EO in the DT and DTP mouse skin on keratinocyte proliferation and papilloma formation.

In an open-label phase 2 study using a combination of BRAF inhibitor dabrafenib and MEK inhibitor trametinib in patients, the rate of skin lesions was not significantly reduced although a slight decrease in proliferative skin lesions was observed [28]. A previous study revealed that tumor multiplicity and incidence of skin tumors in DT-induced two-stage skin carcinogenesis accelerated by BRAF inhibitor was decreased when a COX-2 inhibitor celecoxib was orally administrated [29]. Our current data show that topical application of KWM-EO attenuated the formation of cutaneous papilloma in mice induced by DMBA/TPA or by DMBA/TPA/PLX4032. The paradoxical MAPK activation induced by PLX4032 in vitro in PDV keratinocytes and in skin of DT and DTP mice was suppressed by KWM-EO. Taken together, the results of this study demonstrate the novel chemopreventive activity of the essential oil of *Mentha aquatica*
*var. citrata Kenting Water Mint* which can be potentially used in preventing BRAF inhibitor drug-induced cutaneous side-effects in cancer patients.

## 4. Materials and Methods 

### 4.1. Mint Cultivation and Distillation of Essential Oils

A variety of *Mentha aquatica* (Lamiaceae), named *M. aquatica var. Kenting Water Mint* was cultivated in an experimental field at the Taichung District Agricultural Research and Extension Station, Taichung, Taiwan for 2 years. Mature shoots were harvested and subjected to water vapor distillation to collect essential oils. Two kilograms of fresh shoots were distilled with 4 L of water. Mint essential oil was evaporated, passed through a condenser then the oil and hydrosol were collected with a separating funnel. After 1 L of the hydrosol/essential oils were collected, the distillation ended. The hydrosol and essential oil were then separately collected for use in the following experiments. The mint essential oils were stored at −20 °C in sealed vials. Essential oils used in in vitro cell-based assays were diluted into different concentrations with DMSO and those used in in vivo animal studies were diluted in acetone.

### 4.2. Chemical Profiling of KWM-EO Composition by GC×GC−TOF MS

The samples were analyzed using LECO Pegasus 4D GC×GC−TOF MS (St Joseph, MI, USA). The first dimension capillary column was Restek Rtx-5MS (30 m × 0.25 mm × 0.25 μm) and the second capillary column was Restek Rtx-200 (2 m × 0.25 mm × 0.25 μm). The GC temperature program was set as follows: injection temperature: 280 °C; oven temperature: 40 °C maintained for 1 min, and increased at a rate of 10 °C/min to 310 °C and held constant for 8 min. The helium flow rate was set at 1 mL/min. The mass spectrometry temperature was set at 320 °C. The ion source temperature was 200 °C, and the analysis mass range was 50-800 m/z. KWM-EO was ran in hexane with a dilution of 1 mg/mL. Hexadecane solution, 64.25 μg/mL, was used as an internal standard to monitor the shift of retention time. Compounds were identified by matching the mass spectra fragmentation patterns, and the results were compared with LECO/Fiehn and Wiley Registry 9th Edition mass spectral library and NIST. Linear Kovats index of n-alkanes (C_7_-C_40_, C_7_-C_30_) were calculated for each compound and compared with the literature to identify the compound ID [30].

### 4.3. Cell Lines and Cell Culture

PDV cells, which harbor the *HRAS^Q61L^* mutation were obtained from CLS Cell Lines Service (Eppelheim, Germany). Cells were cultured at 37 °C in DMEM supplemented with 10% FBS, containing 100 units/mL penicillin and 100 μg/mL streptomycin in a humidified 5% CO_2_ incubator. Cells were used within 10 passages for this study.

### 4.4. Measurement of Cell Viability

Cells (5 × 10^3^ cells/well in 96-well plates) were treated with vehicle (0.5% DMSO) or 20, 40, 60, 80, and 100 μg/mL KWM-EO for 24 h. Cell viability was determined by 3-(4,5-Dimethylthiazol-2-yl)-2,5-diphenyl tetrazolium bromide (MTT)-based colorimetric assays according to Scudiero et al. [31]. The viability of the cells treated with vehicle-only was defined as 100% viable. The viability of the cells after treatment with KWM-EO was calculated using the following formula: $cell\;viability\;\left(\%\right)=\left[OD_{570\;\left(treated\;cells\right)}÷OD_{570\;\left(vehicle\;control\right)}\right]\times 100$ empty. The data are presented by three independent experiments with six replicates per experiment.

### 4.5. Colony Formation Assay

Colony formation was obtained by growing PDV cells (250 cells/well in 24-well plates) treated with 10 and 40 μg/mL KWM-EO in the presence or absence of 0.5 μM PLX4032 for 6 days. The culture medium was refreshed once on day 3. Cells were fixed with chilled methanol and stained with 0.1% crystal violet. Cells retaining crystal violet were dissolved with 20% acetic acid and quantified by measuring absorbance at 595 nm [32]. The data are presented by three independent experiments with three replicates per experiment.

### 4.6. Cell Invasion Assay

The cell invasion assay was performed by Millicell Cell Culture Inserts (Merck Millipore, United States). For invasion assay, 100 μL Matrigel (300 μg/mL) was applied to an 8-mm polycarbonate membrane filter and incubated in 37 °C for 2 h. PDV cells (5 × 10^4^) were seeded to Matrigel-coated filters in 200 μL of serum-free medium in triplicate for 16 h. The bottom chamber of the apparatus contained 1 mL medium with 10% FBS as a chemoattractant and 50 and 75 μg/mL KWM-EO, in the presence or absence of 2 μM PLX4032. Cells were allowed to migrate for 24 h at 37 °C. After incubation for 24 h, the non-migrated cells on the apical side of the membrane were removed with cotton swabs. The migrated cells on the basal side of the membrane were fixed with cold 100% methanol for 20 min and washed 3 times with PBS. The cells were stained with 0.1% crystal violet and then washed with PBS to remove extra dye solution. Images were captured using a reverse-phase microscope (Zeiss Axiovert 200M). Cells retaining crystal violet were dissolved with 20% acetic acid and quantified by measuring absorbance at 595 nm. The data are presented by three independent experiments with three replicates per experiment.

### 4.7. Wound Healing Assay

The wound healing assay was performed by using Culture-Insert (ibidi GmbH, Germany). Culture-Inserts were inserted in 24-well plates before cells were seeded. PDV cells were seeded in Culture-Inserts at a density of 5 × 10^5^ cells/mL in 70 μL medium. After 16 h, Culture-Inserts were removed which created two cell-free gaps of 500 ± 50 μm. Undetached cells were washed away by PBS, then the remaining attached cells were immersed in 1 mL medium with 50 μg/mL KWM-EO, in the presence or absence of 2 μM PLX4032. Cell migration was observed using a reverse-phase microscope (Zeiss Observer D1) every 6 h. The data are presented by three independent experiments with three replicates per experiment.

### 4.8. Cell-Cycle Analysis

PDV cells were seeded in 6-well plates at a density of 1 × 10^5^ cells/well with respective medium containing 10% FBS for 16 h. To synchronize the cell-cycle, cells were washed with PBS and incubated with fresh medium containing 5% FBS for 8 h, followed by washing with PBS and incubation with fresh medium containing 0.5% FBS for 24 h. The synchronized PDV cells were then treated with 75 μg/mL KWM-EO, 0.5 μM PLX4032, and 0.5 μM PLX4032+75 μg/mL KWM-EO in the medium containing 10% FBS for 12 and 24 h. Both adherent and floating cells were collected, washed with PBS, and fixed with 500 μL ice-cold 70% ethanol overnight at 4 °C. Cells were stained with 500 μL propidium iodide (PI) solution, which contained 20 μg/mL PI, 20 μg/mL RNase A, 0.1% Triton X-100 for 30 min at room temperature in the dark and then analyzed by flow cytometry (Flow cytometry BD Accuri C6, United States).

### 4.9. Apoptosis Assay

Cells were seeded in 6-well plates at a density of 1.5 × 10^5^ cells/well for 16 h and then treated with 75 μg/mL KWM-EO, 0.5 μM PLX4032, and 0.5 μM PLX4032+75 μg/mL KWM-EO. After 24 h, both adherent and floating cells were collected and washed with PBS. Apoptotic cells were analyzed by using FITC Annexin V Apoptosis Detection Kit (BD Bioscience, United States) according to the manufacturer’s instructions. 

### 4.10. Western Blot Analysis

Cells were treated with KWM-EO at the indicated concentrations in the presence or absence of PLX4032 and lysed in RIPA lysis buffer. Protein concentrations were measured by *DC* protein assay (Bio-Rad, United States). Western blotting was performed as described by Shyur et al. [33]. Primary antibodies ERK 1, cyclin B1, p-cdc2 p34, p-cdc25C, and PARP-1 were purchased from Santa Cruz (Texas, United States). Antibodies phospho-p44/42 MAPK (Erk1/2), MEK1/2, and phospho-MEK1/2 were purchased from Cell Signaling Technology (Massachusetts, United States). Caspase 3 antibody was purchased from GeneTex (Texas, United States). 

### 4.11. Two-Stage Skin Carcinogenesis Study

Female FVB/NJNarl mice (5–6 weeks old) were purchased from the National Laboratory Animal Center (Taipei, Taiwan) and bred in the Laboratory Animal Core Facility (Agricultural Biotechnology Research Center, Academia Sinica, Taiwan). Animals were given a standard laboratory diet and distilled H_2_O *ad libitum* and kept on a 12-h light/dark cycle at 22 ± 2 °C with humidity 55 ± 5%. All experimental protocols were approved by the Institutional Animal Care and Utilization Committee (IACUC: Protocol #18-08-1221), Academia Sinica, Taiwan. Mice were randomized and had their back hair shaved three days before topical application of 25 μg DMBA in 200 μL acetone. The first week after tumor initiation, 4 μg of TPA in 200 μL acetone was topically applied twice a week to the shaved dorsal skin for 12 weeks [5]. Mice were treated with the indicated concentration of KWM-EO (in 200 μL acetone) twice a week by topical application the day after TPA treatment for 12 weeks (Appendix A). Tumor size of more than 1 mm diameter was counted every week.

### 4.12. Histopathological and Immunohistochemical Analysis

Tissues were fixed with 10% formalin, hydrated, and embedded in paraffin. Tissue sections were cut at 4 μm thickness, then deparaffinized following rehydration in a descendant ethanol bath. H&E staining, immunohistochemistry, and immunofluorescent staining followed the previously published protocols [30]. An upright microscope (Carl Zeiss Axio Imager, Z1) was used to observe the expression of targeted proteins. Primary antibodies cytokeratin 14 and CD163 were purchased from Proteintech (Illinois, United States). Ki67 and neutrophil elastase were purchased from Abcam (Cambridge, United Kingdom). Antibody against COX-2 was purchased from Cayman (Michigan, United States). Antibody against F4/80 was purchased from Biolegend (California, United States). Antibody against iNOS was purchased from BD transduction Laboratories (California, United States).

### 4.13. Statistical Analysis

All the data are expressed as mean ± standard deviation (SD). Statistical analyses were conducted by the Predictive Analysis Suite Workstation (PASW Statistics, United States), and the significant difference between different treatment groups was determined by analysis of variance (ANOVA). *P* values of less than 0.05 were considered statistically significant. 

## 5. Conclusions

This study is the first to prove that KWM-EO has potential for prevention of chemically induced two-stage skin carcinogenesis. Topical application of KWM-EO significantly attenuated the number of papillomas in DMBA-initiated and TPA-promoted mouse skin, with or without co-stimulation with PLX4032. KWM-EO suppressed epidermal hyperplasia and over proliferation of keratinocytes in DMBA/TPA and DMBA/TPA/PLX4032 mice. Notably, KWM-EO treatment diminished MAPK pathway reactivation, pro-inflammatory immune cell infiltration, and COX-2 expression in both DMBA/TPA and DMBA/TPA/PLX4032 mouse skin tissues. Overall, the results in this study provide strong support for the development of KWM-EO into chemopreventive agents for squamous cell carcinoma patients or cancer patients taking BRAF inhibitor therapy.

## Figures and Tables

**Figure 1 molecules-24-02344-f001:**
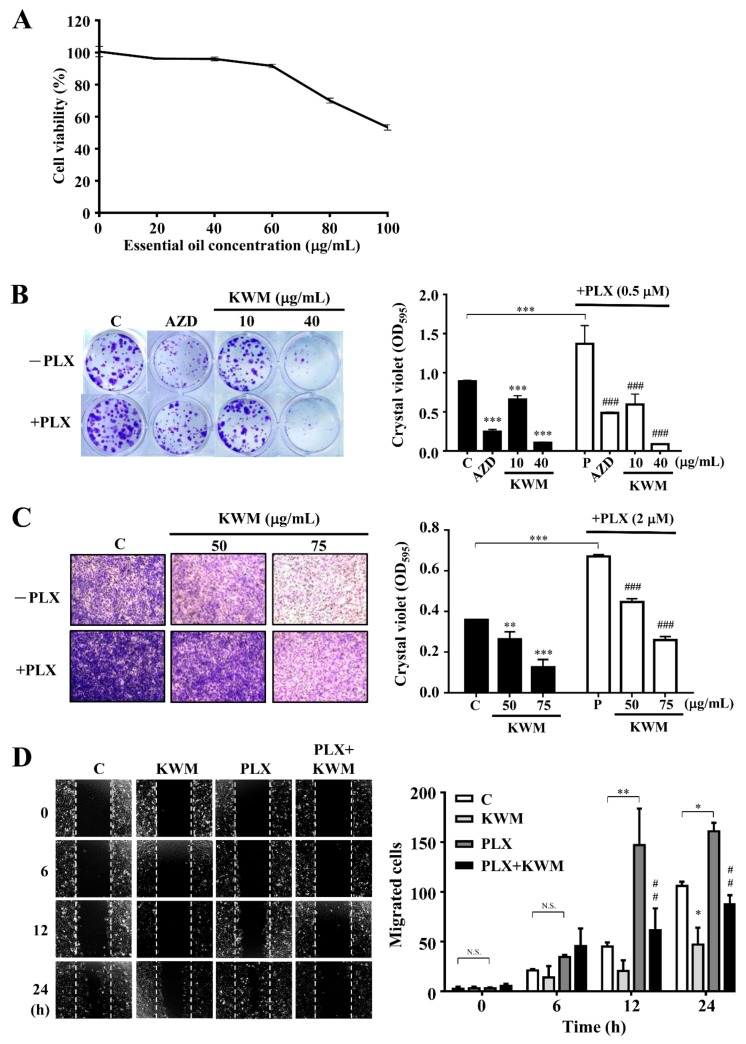
Effect of *Mentha aquatica var. Kenting Water Mint* essential oil (KWM-EO) on PDV cells. (**A**) PDV cells were treated with vehicle or the indicated concentrations of KWM-EO for 24 h. Cell viability (%) was determined by MTT assay. (**B**) PDV cells were incubated with KWM-EO in the presence or absence of 0.5 μM PLX4032 for 6 days, and colony formation was detected by staining cells with crystal violet. (**C**) PDV cells were seeded in Matrigel coated–transwell inserts and incubated with vehicle or KWM-EO in the presence or absence of 2 μM PLX4032 for 24 h. The invasive cells were stained with crystal violet. (**D**) PDV cell migratory ability was examined by wound healing assay. Cells were treated with vehicle or 50 μg/mL KWM-EO in the presence or absence of 2 μM PLX4032, and observed after 0, 6, 12, 24 h. Vehicle controls (C) were obtained from cells treated with 0.5% DMSO. The absorbance at 595 nm was obtained by dissolving crystal violet with 20% acetic acid. The data are representative of three independent experiments and are expressed as mean ± SD. Representative images are shown. *P** < 0.05, *P*** < 0.01, *P**** < 0.001 compared to vehicle control; *P*^##^ < 0.01, *P*^###^ < 0.001 compared to the PLX4032-treated group (ANOVA). AZD: AZD6244 (MEK inhibitor)

**Figure 2 molecules-24-02344-f002:**
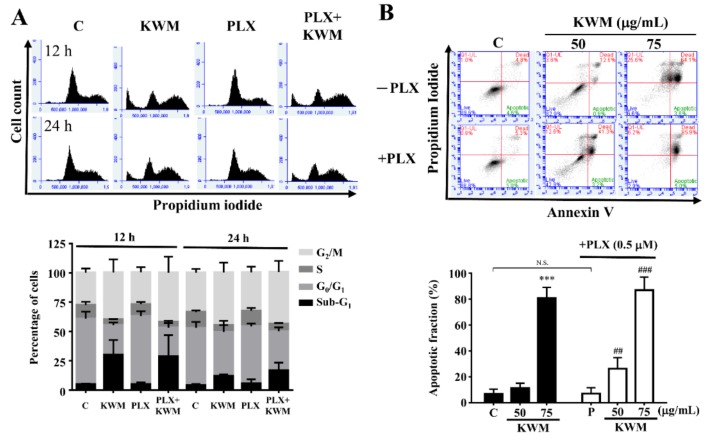
Effect of KWM-EO on cell-cycle and apoptosis in PDV cells. (**A**) PDV cells were exposed to vehicle or 75 μg/mL KWM-EO in the presence or absence of 0.5 μM PLX4032 for 12 or 24 h, then the cell-cycle was analyzed by flow cytometry. (**B**) PDV cells were treated with vehicle or 75 μg/mL KWM-EO in the presence or absence of 0.5 μM PLX4032 for 24 h. Cells were then stained with annexin V and propidium iodide, and the cell apoptosis was detected by flow cytometry. (**C**) PDV cells were treated with 75 μg/mL KWM-EO in the presence or absence of 0.5 μM PLX4032 for 24 h before lysis. The cell lysates were subjected to Western blotting against cell-cycle-related proteins, including p-cdc2 (Thr161), p-cdc25C, and cyclin B1. (**D**) The expression level of apoptosis-related proteins in PDV cells treated with 75 μg/mL KWM-EO in the presence or absence of 0.5 μM PLX4032 for 6 h was examined by Western blotting against PARP-1 and caspase 3. (**E**) Western blotting analysis of MAPK signaling-related proteins (p-ERK, ERK, p-MEK, MEK) in PDV cells treated with 75 μg/mL KWM-EO in the presence or absence of 0.5 μM PLX4032 for 24 h. Actin was used as an internal control in the experiment. Vehicle controls (**C**) were obtained from cells treated with 0.5% DMSO. The data are representative of three independent experiments and are expressed as mean ± SD. N.S. means non significance; *P**** < 0.05 compared to vehicle control; *P*^##^ < 0.01, *P*^###^ < 0.001 compared to PLX4032-treated group (ANOVA).

**Figure 3 molecules-24-02344-f003:**
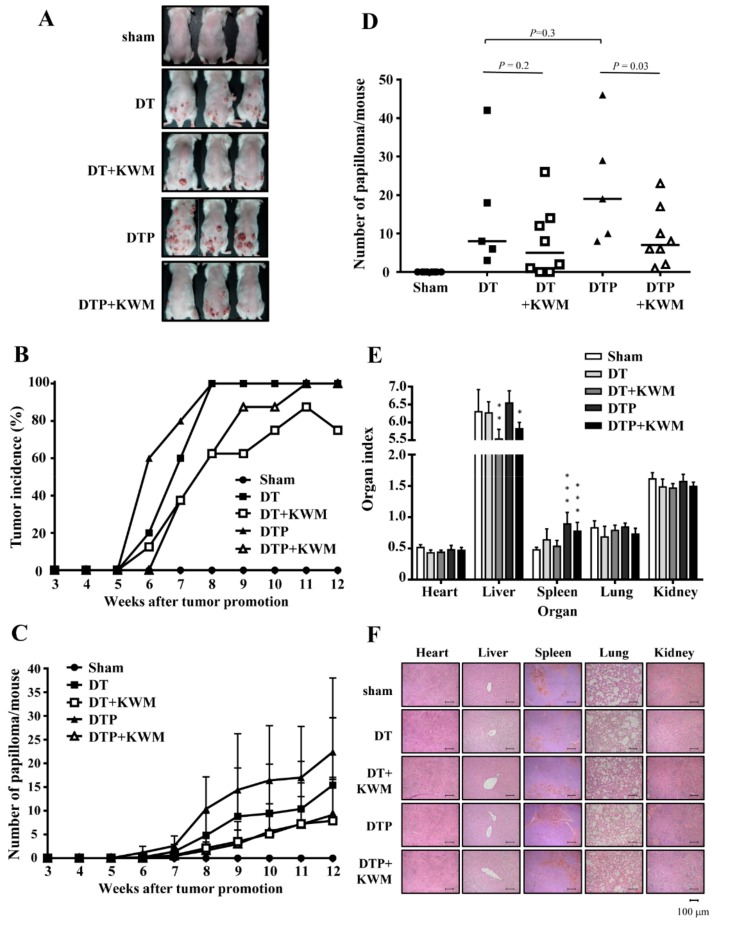
Effect of KWM-EO on two-stage skin carcinogenesis in mice. (**A**) Representative images of mice from each group at week 12 are shown. Tumor incidence (**B**) and mean number of papillomas (**C**) per group during the experimental period are calculated. (**D**) Papilloma numbers per mouse at week 12 are shown in the dot histogram. (**E**) Organ weights were recorded after mice were sacrificed at week 12. Organ index was calculated by the following formula: $organ\;index=\left(organ\;weight÷body\;weight\right)\times 100$ empty. *P** < 0.05, *P*** < 0.01, *P**** < 0.001 compared to sham group (ANOVA). (**F**) Organ tissues were detected by H&E staining. The DT and DTP groups consisted of 5 mice each. Sham, the DT+KWM, and DTP+KWM groups consisted of 8 mice each. The data are presented as mean ± SD. D: DMBA, T: TPA, P: PLX4032. Scale bar: 100 μm.

**Figure 4 molecules-24-02344-f004:**
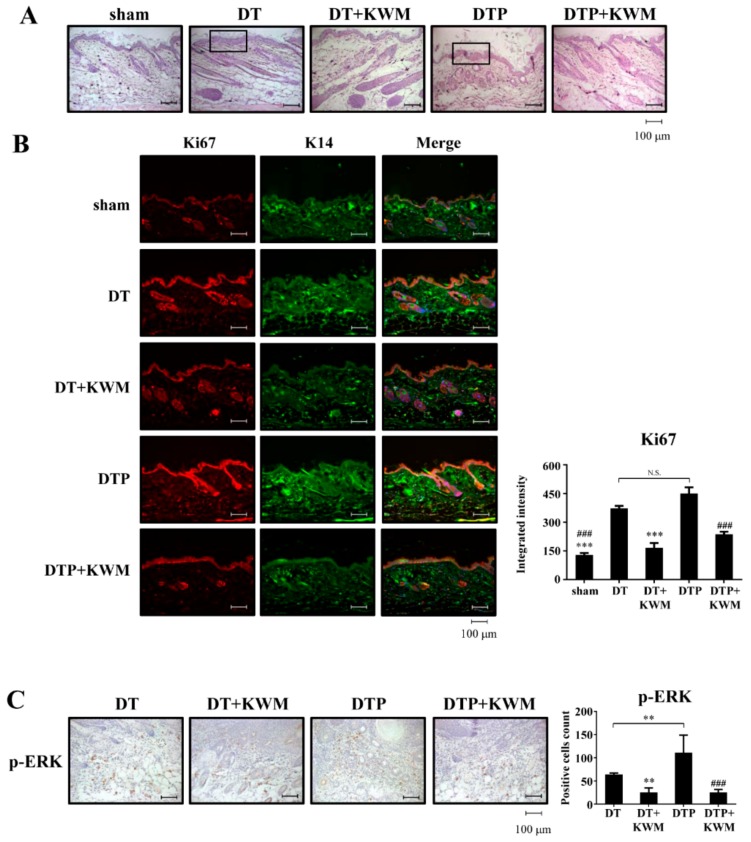
Effect of KWM-EO on skin and papilloma tissue from mice. (**A**) Skin morphology was examined by H&E staining. (**B**) Abnormal epidermal proliferation was detected by immunofluorescent staining of ki67 (red). Basal keratinocytes were stained with K14 (green), and nuclei were stained with DAPI (blue). (**C**) Histological images of papilloma indicated the paradoxical activation of p-ERK. Representative images are shown. The data are representative of three independent experiments and are expressed as mean ± SD. N.S. means non significance; *P*** < 0.01, *P**** < 0.001 compared to DT group; *P*^###^ < 0.001 compared to DTP group (ANOVA). Scale bar: 100 μm.

**Figure 5 molecules-24-02344-f005:**
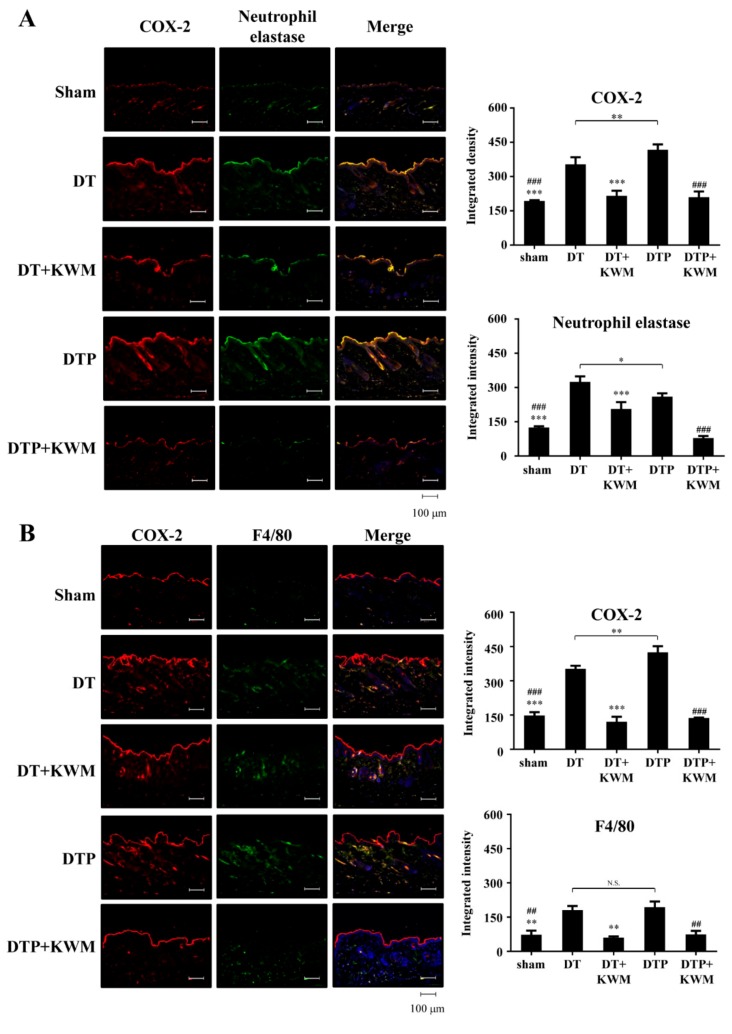
Effect of KWM-EO on the inflammatory immune system in skin tissue from the mice two-stage skin carcinogenesis. Immunofluorescent staining of inflammatory mediator COX-2 (red) with neutrophil elastase (green) (**A**) and macrophage marker, F4/80 (green) (**B**). Nuclei were counterstained with DAPI (blue). Representative images are shown. The data are representative of three independent experiments and are expressed as mean ± SD. N.S. means non significance; *P** < 0.05, *P*** < 0.01, *P**** < 0.001 compared to DT group; *P*^##^ < 0.01, *P*^###^ < 0.001 compared to DTP group (ANOVA). Scale bar: 100 μm.

**Table 1 molecules-24-02344-t001:** Chemical constituents of KWM-EO determined by GC×GC-TOF MS.

	Chemical Compound	CAS no.^a^	RT1^b^	RT2^c^	KI_exp_^d^	KI_Lit_^e^	Relative Percentage (%)
1	α-Pinene	7785-70-8	6.27	0.0245	936	929	10.49
2	β-Pinene	127-91-3	7.00	0.0250	981	973	15.41
3	β-Myrcene	123-35-3	7.20	0.0243	992	993	4.86
4	β-Cymene	535-77-3	7.73	0.0253	1027	1031	3.07
5	Eucalyptol	470-82-6	7.93	0.0258	1040	1041	12.87
6	β-Ocimene	3338-55-4	8.07	0.0245	1049	1036	22.18
7	Linalool	78-70-6	8.87	0.0262	1097	1101	0.25
8	Menthone	89-80-5	9.80	0.0325	1160	1154	0.04
9	Menthofuran	494-90-6	9.93	0.0267	1169	1169	0.04
10	Levomenthol	2216-51-5	10.07	0.0277	1178	1172	0.05
11	*p*-Menth-8-en-2-one	5948-4-9	10.60	0.0330	1213	1218	0.07
12	Carveol	1197-07-5	10.73	0.0272	1222	1223	0.13
13	Carvone	6485-40-1	11.13	0.0327	1251	1249	1.59
14	Linalyl acetate	115-95-7	11.20	0.0270	1256	1257	0.08
15	Dihydroedulan I	63335-66-0	11.87	0.0258	1302	1292	0.28
16	β-Bourbonene	5208-59-3	13.13	0.0250	1396	1386	0.78
17	Caryophyllene	87-44-5	13.60	0.0258	1434	1431	2.80
18	Humulene	6753-98-6	14.00	0.0318	1466	1465	0.29
19	Ethyl 4-ethoxybenzoate	23676-09-7	14.73	0.0302	1524	1521	2.37
20	Viridiflorol	552-02-3	15.73	0.0272	1608	1603	3.47
**Monoterpene hydrocarbons identified**							56.01
**Oxygenated monoterpene identified**							15.86
**Sesquiterpene hydrocarbon identified**							3.87
**Oxygenated sesquiterpene identified**							3.47
**Other**							2.56
**Identified components**							81.86

^a^ Chemical abstracts service registry number; ^b^ Retention time of the first column; ^c^ Retention time of the second column; ^d^ KI_exp_ = Kovats indices, retention indices relative to C_7_–C_30_ n-alkanes based on the retention time of components separated by the 1st dimension Rtx-5MS column; ^e^ KI_Lit_: Retention indices reported in the literature.

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
