# Peer review of "Essential Oil of Mentha aquatica var. Kenting Water Mint Suppresses Two-Stage Skin Carcinogenesis Accelerated by BRAF Inhibitor Vemurafenib"

_molecules, 2019, doi:10.3390/molecules24122344_

Round 1

Reviewer 1 Report

The study evaluates the bioactivity of Mentha aquatica var. Kenting Water Mint essential oil (KWM-EO) against PDV cells, mouse keratinocytes bearing HRASQ61L mutation, and its effect on inhibiting papilloma formation in a two-stage skin carcinogenesis mouse model with or without PLX4032 co-treatment. The results are interesting and the conclusion are justified.

Minor point

Could be authors provide some conjecture, on the basis of previous literature, regarding the active compounds that might be involved in the specific effects that were observed?

Author Response

Reponses to the reviewer comments by manuscript (#molecules-526639)

 Response to the Reviewer 1 comments

Minor Point: Could be authors provide some conjecture, on the basis of previous literature, regarding the active compounds that might be involved in the specific effects that were observed?

Response: We thank the reviewer for his/her constructive suggestion. We have done literature searches and observed few of the compounds present in our target mint essential oil KWM-EO have been suggested for their bioactivity. For example, a previous review article published by Pandey et al [24] summarized that EOs of some Ocimum species exhibited anti-inflammatory and anti-cancerous properties which contain pinene, b-ocimene and linalool, the chemical constituents present in KWM-EO. a-Pinene present in KWM-EO by 10.49% was reported to induce cell apoptosis and disrupt mitochondrial potential in B16F10 cells and it also effectively reduced melanoma lung metastasis [25]. b-Caryophyllene accounted for 2.8% in KWM-EO was a major compound in the EO of P. missionis. Pavithra et al. demonstrated that EO from P. missionis induced cell death through intrinsic mitochondrial and extrinsic apoptotic pathway in A431 and HaCaT cells [26]. The results from these studies might potentially support the role of constituent compounds in the KWM-EO exhibiting anti-inflammatory and chemopreventive activities against two-stage skin carcinogenesis.  We have added these information in the Discussion section of the revised manuscript on page 11. The References section has been alos updated by adding the three new references.

[24] Pandey, A.K.; Singh, P.; Tripathi, N.N. Chemistry and bioactivities of essential oils of some Ocimum species: an overview. Asian Pac J Trop Biomed 2014;4:682-94

[25] Matsuo, A.L.; Figueiredo, C.R.; Arruda, D.C. et al. a-Pinene isolated from Schinus terebinthifolius Raddi (Anacardiaceae) induces apoptosis and confers antimetastatic protection in a melanoma model. Biochem Biophys Re. Commu. 2011;411:449-54

[26] Pavithra, P.S.; Mehta, A.; Verma, R.S. Induction of apoptosis by essential oil from P. missionis in skin epidermoid cancer cells. Phytomedicine 2018;50:184-95

Reviewer 2 Report

Presented study entitled: "Essential oil of Mentha aquatica var. Kenting Water Mint suppresses two-stage skin carcinogenesis accelerated by BRAF inhibitor vemurafenib" reflect potential way to answering to the high demand question about the treatment of skin cancer.

Article is scientificaly in high level.

As a reviewer I have some suggestions to the improvement of the presentation of article:

Table 1: It is ok, when the relative percentage are expressed with only one decimal number –in this case menthon and menthofuran could be omited or mark it as a traces (t). Also unify the numbers of decimal numbers at KI1 –and KI2 –lets all numbers are expressed in the same way.

Lines 89, 219-220: Avoid citation in the part of Results. Generalize the sentences without using citation.

Line 91-92: there is not clear expression of treatment of different concentrations of EO - how many and which concentrations of EO were used?. According to Fig.1A it seems that until concentration 60 mg/ml the viability of cells was more than 80 %. Increased concentration decrease viability.. Could you please rewrite it more exact?

Figures-all which may concern: I would recommend to change the expression of „V“ (as vehicle control) to the „C“ which is more general and faster to recognize for all readers.

Lines 119-121: This sentences should be moved to the methodology section.

Line 290: Is the name of the analytical equipment correct? Pegasus 4D 2D GC×GCTOF MS? If the analysis was 2 D.

Line 307: 4.4 Measurement of cell viability / 4.5 Colony formation assay / 4.6 Cell invasion assay

Please add which and how many different concentrations of  KWM-EO were used for treatment. Is it possible to explain in how many holes (or plates) were used for the different concentrations of KWM-EO – as we can imagine how many replications were done.. Also explain, why PLX4032 (PLX) were added, along which each concentraion of EO? Or selected- it is not clear in expression of results, when it is missing in Method.

Line 389: correct KEM-EO to KWM-EO

Improve discussion by using more comparable studies with different Metha EO as well as the effect of different EO concentrations.

Author Response

Reponses to the reviewer comments by manuscript (#molecules-526639)

Reponses to the reviewer comments by manuscript (#molecules-526639)

Response to the Reviewer 2 comments

Point 1: Table 1: It is ok, when the relative percentage are expressed with only one decimal number –in this case menthon and menthofuran could be omited or mark it as a traces (t). Also unify the numbers of decimal numbers at KI1 –and KI2 –lets all numbers are expressed in the same way.

Response: We thank the reviewer for the suggestion. In the revised manuscript, we have uniformed the expression of the numbers under the same category in Table 1 with the same decimal numbers.  We have also decided to keep two decimal number for the relative percentage of all the compounds detected in KWM-EO that we believe would be more completely present all the detected chemical constituents in the essential oil.

Point 2: Lines 89, 219-220: Avoid citation in the part of Results. Generalize the sentences without using citation.

Response: We have modified the sentence as mentioned and also the sentences on lines 89-90, 219-220 have been removed for the citation.

Point 3: Line 91-92: there is not clear expression of treatment of different concentrations of EO - how many and which concentrations of EO were used? According to Fig.1A it seems that until concentration 60 mg/ml the viability of cells was more than 80 %. Increased concentration decrease viability. Could you please rewrite it more exact?

Response: We have added the used concentration rang of KWM-EO (0-100 mg/mL) on page 3, line 91-92. The five concentrations of KEM-EO used in the experiment (20, 40, 60, 80, 100 mg/mL) were added in the Materials and Methods section on page 12, line 322-323.

Point 4: Figures-all which may concern: I would recommend to change the expression of „V“ (as vehicle control) to the „C“ which is more general and faster to recognize for all readers.

Response: We have replaced V by C in Figures 1 and 2 of the revised manuscript.

Point 5: Lines 119-121: This sentences should be moved to the methodology section.

Response: We have moved the sentence to line 363-365 on page13.

Point 6: Line 290: Is the name of the analytical equipment correct? Pegasus 4D 2D GC×GCTOF MS? If the analysis was 2 D.

Response: We apologize for the confusing of the wordings that we made. Actually Pegasus 4D is part of the name of the equipment. We have modified the sentence as “The samples were analyzed using LECO Pegasus 4D GC×GC-TOF MS (St Joseph, MI, USA).” in the revised manuscript.

Point 7: Line 307: 4.4 Measurement of cell viability / 4.5 Colony formation assay / 4.6 Cell invasion assay

Please add which and how many different concentrations of KWM-EO were used for treatment. Is it possible to explain in how many holes (or plates) were used for the different concentrations of KWM-EO – as we can imagine how many replications were done.. Also explain, why PLX4032 (PLX) were added, along which each concentraion of EO? Or selected- it is not clear in expression of results, when it is missing in Method.

Response: The concentrations and the number of technical replicates relevant to the holes used in the experiment are now updated in the sub-sections 4.4 Measurement of cell viability, 4.5 Colony formation assay, 4.6 Cell invasion assay, and 4.7 Wound healing assay in the revised manuscript. A parallel set of experiment adding extra PLX4032 as a stimulator in the cells was carried out at the same time with the identical KWM-EO treatments. These have been also described in the subsections mentioned above.

Point 8: Line 389: correct KEM-EO to KWM-EO

Response: We have corrected the typo accordingly. Thank you.

Point 9: Improve discussion by using more comparable studies with different Metha EO as well as the effect of different EO concentrations.

Response: We thank the reviewer for the constructive suggestion. We have added new sentences and one new reference in the Discussion on page 10 which is quoted below.

“The ethanolic extract of Mentha×piperita L., a cross species of watermint and spearmint, at 50 and 100 µg/mL, suppressed LPS-induced nitric oxide production in macrophages by 18.85% and 41.88% inhibition, respectively [22]. Anti-cancer cell activity has been also reported for mint EOs. For example, Mentha×piperita L. extract showed more potent activity against proliferative activity of human MDA-MB-231 breast cancer cells (cell inhibition ratio = 46.53% at 150 mg/mL) than human A375 melanoma cells (cell inhibition ratio = 25.08% at 150 mg/mL) [14].”

In addition, we have also added a new paragraph in the Discussion section (line 261-269) which is also quoted below:

A previous review article published by Pandey et al. [24] summarized that EOs of some Ocimum species exhibited anti-inflammatory and anti-cancerous properties which contain pinene, b-ocimene and linalool, the chemical constituents present in KWM-EO. a-Pinene present in KWM-EO by 10.49% was reported to induce cell apoptosis and disrupt mitochondrial potential in B16F10 cells and it also effectively reduced melanoma lung metastasis [25]. b-Caryophyllene accounted for 2.8% in KWM-EO was a major compound in the EO of P. missionis. Pavithra et al. demonstrated that EO from P. missionis induced cell death through intrinsic mitochondrial and extrinsic apoptotic pathway in A431 and HaCaT cells [26]. The results from these studies might potentially support the role of constituent compounds in the KWM-EO exhibiting anti-inflammatory and chemopreventive activities against two-stage skin carcinogenesis.”